

# Nrf2 pathway activation promotes the expression of genes related to glutathione metabolism in alcohol-exposed astrocytes

Congyan Li[1,*], Jingxin Fan[2,*], Guangtao Sun[1], Huiying Zhao[3], Xiaogang Zhong[4,5], Xinyan Huang[6], Xiaofeng Zhu[7] and Xunzhong Qi[1]

[1] Department of Neurology, The First Affiliated Hospital of Jiamusi University, Jiamusi, China
[2] Jiamusi University, Jiamusi, People's Republic of China
[3] Department of Neurology, Yichun Forestry Administration Central Hospital, Yichun, China
[4] College of Basic Medicine, Chongqing Medical University, Chongqing, China
[5] NHC Key Laboratory of Diagnosis and Treatment on Brain Functional Diseases, The First Affiliated Hospital of Chongqing Medical University, Chongqing, China
[6] The Second Affiliated Hospital of Jiamusi University, Jiamusi, China
[7] Mudanjiang Medical College, Mudanjiang, China
* These authors contributed equally to this work.

Corresponding authors
Xiaofeng Zhu,
zhuxiaofeng_mdj@163.com
Xunzhong Qi, 617han@163.com

## ABSTRACT

**Introduction:** Oxidative and antioxidant pathways play essential roles in the development of alcohol-induced brain injury. The Nrf2 pathway is an endogenous antioxidant response pathway, but there has been little research on the role of Nrf2 in alcohol-related diseases. Thus, we examined the effects of alcohol and an Nrf2 agonist (TBHQ) on astrocyte function, mRNA expression, and metabolite content to further explore the protective mechanisms of Nrf2 agonists in astrocytes following alcohol exposure.

**Methods:** CTX TNA2 astrocytes were cultured with alcohol and TBHQ and then subjected to transcriptome sequencing, LC-MS/MS analysis, quantitative reverse transcription polymerase chain reaction (qRT-PCR), and malondialdehyde (MDA) and superoxide dismutase (SOD) activity assays.

**Results:** Alcohol exposure significantly increased malondialdehyde (MDA) levels while decreasing superoxide dismutase (SOD) levels in astrocytes. Treatment with TBHQ effectively reversed these effects, demonstrating its protective role against oxidative stress induced by alcohol. Transcriptome sequencing and qRT-PCR analysis revealed that TBHQ specifically upregulates genes involved in glutathione metabolism, including a notable increase in the expression of the glutathione S-transferase A5 (GSTA5) gene, which was suppressed by alcohol exposure. Additionally, metabolomic analysis showed that TBHQ regulates key components of ether lipid metabolism in alcohol-exposed astrocytes, with significant reductions in the levels of lysophosphatidylcholine (18:0) (LysoPC (18:0)) and 2-acetyl-1-alkyl-sn-glycero-3-phosphocholine, both of which are critical markers in the ether lipid metabolic pathway.

**Discussion:** The findings underscore the role of TBHQ as an Nrf2 agonist in mitigating alcohol-induced oxidative damage in astrocytes by modulating glutathione metabolism and ether lipid metabolism. The regulation of GSTA5 gene expression emerges as a key mechanism through which Nrf2 agonists confer neuroprotection against oxidative stress and lipid oxidation. These insights pave the

way for potential therapeutic strategies targeting the Nrf2 pathway to protect astrocytes from alcohol-induced damage.

# INTRODUCTION

Continuous alcohol intake can lead to a decreased brain volume and the loss of nerve cells in the cortex, especially in the prefrontal cortex (*Crews & Vetreno, 2014*). Additionally, it can lead to a variety of neuropathological lesions, such as cognitive impairment and dementia (*Qin & Crews, 2012*). Oxidation and inflammation play important roles in this process. Astrocytes are essential targets of alcohol in brain tissue (*de la Monte & Kril, 2014*), and the number of astrocytes is greatly modulated by alcohol. Prior studies illustrated that lower astrocyte density is seen in the dorsolateral and orbitofrontal prefrontal cortex and hippocampus of human alcoholics (*Adermark & Bowers, 2016*) and the number of glial fibrillary acidic protein-positive astrocytes in the prelimbic prefrontal cortex is reduced in rats with alcohol preference (*Miguel-Hidalgo, 2005*). In the central nervous system, astrocytes express Toll-like receptors, which are clearly associated with alcohol-induced inflammation and oxidative stress. Oxidative and antioxidant pathways play essential roles in the process of alcohol-induced brain injury. Previous studies revealed that sustained alcohol intake increases nitrogen oxide levels in the brain and promotes the production of reactive oxygen species, thereby stimulating an oxidative stress response (*Qin & Crews, 2012*). The Nrf2 pathway is an endogenous antioxidant response pathway that acts as an anti-oxidative stressor by modulating the secretion of antioxidant enzymes in combination with the antioxidant response element (ARE) (*Sanghvi et al., 2019*). Research highlights that activation of NRF2 can significantly improve mitochondrial functionality and alleviate neuroinflammation, which are essential in both preventing and repairing brain damage from alcohol (*Zhao et al., 2022*). Furthermore, Octreotide significantly attenuated chronic ethanol-induced neuropathic pain and it also restored the levels of Nrf2 (*Jiang & Wei, 2021*). This underscores the potential of Nrf2 activation not just in countering oxidative damage but also as a viable approach for addressing the neurodegenerative impacts of alcohol. However, the mechanism of alcohol-induced astrocyte injury is still poorly understood.

To further analyze the role and mechanisms of the Nrf2 pathway in alcohol-related diseases, the Nrf2 pathway agonist tert-butylhydroquinone (TBHQ) was used to intervene in alcohol-exposed astrocytes. TBHQ is a major metabolite of butylated hydroxyanisole and induces an antioxidant response through the redox-sensitive transcription factor Nrf2 (*Imhoff & Hansen, 2010*). To observe the effects and mechanisms of alcohol and TBHQ on oxidative stress in astrocytes, transcriptome sequencing and non-targeted metabolite analysis were used to investigate the regulatory mechanisms of astrocyte function following exposure to alcohol and TBHQ. This study provides experimental evidence to elucidate the mechanisms of astrocytes injury caused by persistent alcohol intake and

theoretical support for the application of Nrf2 pathway-targeting drugs in ethanol-related disease.

## MATERIALS AND METHODS

### Cell culture and treatments

CTX TNA2 astrocytes were obtained from ATCC Corporation and divided into three groups: control, TBHQ (TBHQ + alcohol), and alcohol. To investigate the impact of TBHQ on astrocytes exposed to alcohol, the cells in the TBHQ group were treated with 40 μM TBHQ for 24 h, following established protocols (*Zhang et al., 2017*). Subsequently, the medium was replaced with either normal complete medium (control group) or complete medium containing an alcohol concentration of 75 mM (*Wilhelm et al., 2018*) (0.35% w/v, corresponding to 0.35 g/dl) (alcohol and TBHQ groups). This ethanol concentration is clinically relevant, as it can be found in the blood of individuals after high ethanol intake (*Adachi et al., 1991*), and is within the range of concentrations recommended *in vitro* (*Deitrich & Harris, 1996*). After another 24-h culture period, the cells were collected for subsequent experiments. The levels of malondialdehyde (MDA) and superoxide dismutase (SOD) activities were determined using assay kits according to manufacturer instructions.

### RNA sequencing (RNA-seq) and quantitative real-time polymerase chain reaction (qRT-PCR) analysis

The statistical power of this experimental design, calculated in RNASeqPower is 0.72. There were four samples in each group, and the sequencing depth was 6 G Cleandata. Total RNA was obtained from CTX TNA2 astrocyte samples using TRIzol® reagent. Differential expression analysis was performed using DESeq2, and genes with |fold change| > 1.5 and $P$ adjust < 0.05 were considered significant differentially expressed genes (DEGs). Kyoto Encyclopedia of Genes and Genomes (KEGG) pathways with a false discovery rate smaller than 0.05 were selected.

RNA was isolated from CTX TNA2 astrocyte samples using TRIzol® Reagent (Thermo Fisher Scientific, Waltham, MA, USA) and subsequently reverse-transcribed using the PrimeScript™ RT reagent Kit with gDNA Eraser (Perfect Real Time RR047A) (Takara, Otsu, Japan). The relative expression was calculated using the $2^{-\Delta\Delta Ct}$ method and estimated relative to GAPDH. Data were collected as previously described (*Xunzhong et al., 2023*). The minimum information for publication of quantitative real-time PCR experiments is listed in the Supplemental Material.

### LC-MS/MS and metabonomics data analysis

The samples were subjected to analysis using a UHPLC-Q Exactive HF-X mass spectrometer from Thermo Fisher Scientific, following the previously described protocol (*Zhao et al., 2023*). Metabolites that exhibited a variable importance in projection (VIP) value greater than 1 and a $P$-value less than 0.05 were considered as significant metabolites. The metabonomics data can be accessed at MetaboLights: MTBLS5071.

## Statistical analysis

All qRT-PCR data were analyzed using the SPSS 21.0 software (SPSS Inc.; IBM, Chicago, IL, USA), the differences among the three groups were evaluated using a one-way analysis of variance (ANOVA) followed by a *post hoc* test to determine the least significant difference. A significance level of $P < 0.05$ was considered statistically significant.

# RESULTS

## Alcohol exposure increases MDA levels and decreases SOD levels in astrocytes, which were reversed by TBHQ

In this experiment, we observed the effect of alcohol and TBHQ on the MDA and SOD content of astrocytes. The findings demonstrated a significant increase in MDA levels by 13.7% ($P = 0.014$) and a decrease in SOD levels by 14.7% ($P = 0.048$) in astrocytes following alcohol exposure, compared to control levels (Figs. 1A, 1B). Conversely, after treatment with TBHQ, there was a notable reduction in MDA levels by 18.6% ($P = 0.001$) and an increase in SOD levels by 29.3% ($P = 0.003$) in astrocytes, relative to those exposed to alcohol (Figs. 1A, 1B).These results suggest that alcohol promotes astrocyte oxidation, which can be reversed by TBHQ.

## Data re-analysis revealed that alcohol exposure modulates the expression of metabolism-related genes

We re-analyzed the data for the alcohol and control groups previously described by our team (*Xunzhong et al., 2023*) using different analysis methods and identified 91 DEGs in alcohol-exposed astrocytes compared with control cells, including 30 upregulated and 61 downregulated genes. Metabolism-related pathways were the most annotated pathways in KEGG annotation analysis (Fig. 2A). The most enriched KEGG pathway related to these DEGs was "Arachidonic acid metabolism" (rno00590, $P = 0.0062$, Fig. 2B). Specifically, the first 12 enrichment pathways were all related to metabolism (Fig. 2B). These results suggest that alcohol exposure modulates the expression of genes involved in astrocyte metabolic processes.

## TBHQ modulates the expression of genes involved in astrocyte glutathione metabolism after alcohol exposure

Compared with the alcohol group, 49 DEGs were identified in astrocytes treated with TBHQ, including 40 upregulated and nine downregulated genes (Fig. 3). Metabolism-related pathways were the most annotated pathways in KEGG annotation analysis (Fig. 4A). The most enriched KEGG pathway related to these DEGs was "Glutathione metabolism" (RNO00480, $P < 0.0001$, Fig. 4B). These results suggest that TBHQ modulates the expression of genes involved in glutathione metabolism in alcohol-exposed astrocytes.

## Alcohol exposure decreases glutathione S-transferase A5 (GSTA5) gene expression in astrocytes, which was reversed by TBHQ

Because of the multiple similarities between the control and alcohol groups and the alcohol and TBHQ groups in KEGG analysis, we analyzed the six overlapping DEGs in the control

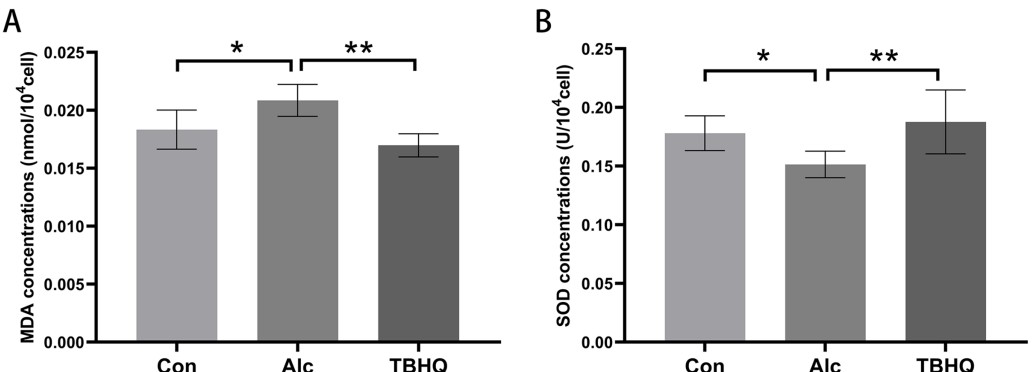

**Figure 1 Alcohol exposure increases MDA levels and decreases SOD levels in astrocytes, and these changes were reversed by TBHQ.** (A) MDA levels in astrocytes after alcohol and TBHQ intervention ($n = 5$). (B) SOD levels in astrocytes after alcohol and TBHQ intervention. ($n = 5$). $^*P < 0.05$; $^{**}P < 0.01$.

and alcohol groups and the alcohol and TBHQ groups *via* KEGG analysis (Figs. 5A, 5C). The most enriched KEGG terms related to these six DEGs were "AGE-RAGE signaling pathway in diabetic complications" (RNO04933, $P = 0.0342$) and "Glutathione metabolism" (RNO00480, $P = 0.0449$, Fig. 5D). One gene related to glutathione metabolism, namely GSTA5, was selected for verification by qRT-PCR. The expression of GSTA5 was consistent with the sequencing results (Figs. 5B, 5C). Compared with the control group, the alcohol group exhibited decreased GSTA5 mRNA expression ($P = 0.039$). Compared with the alcohol group, the TBHQ group displayed increased GSTA5 mRNA expression ($P = 0.002$). Together, these results indicate the accuracy of the RNA-seq results and suggest that glutathione metabolism and GSTA5 mRNA expression play critical roles in regulating astrocyte function after alcohol and TBHQ exposure.

## Effects of TBHQ on alcohol-induced changes in astrocyte metabolomics—PCA and OPLS-DA of samples from the alcohol and TBHQ groups

PCA of the differences between groups demonstrated that the alcohol and TBHQ groups tended to separate under anionic and cationic modes, but the relative concentration within each group was poor. Thus, OPLS-DA was needed to maximize differences between the two groups. The OPLS-DA score chart revealed good intergroup differences and intragroup aggregation, indicating that the model had good explanatory and predictive performance (Figs. 6A, 6B).

In the OPLS-DA model, the significantly altered metabolites were identified by VIP > 1.0 in the OPLS-DA models. In total, 38 different metabolites in the alcohol group astrocyte were significantly altered by TBHQ. Compared with the alcohol group, 20 metabolites were decreased, and 18 metabolites were increased in the TBHQ group (Fig. 7).

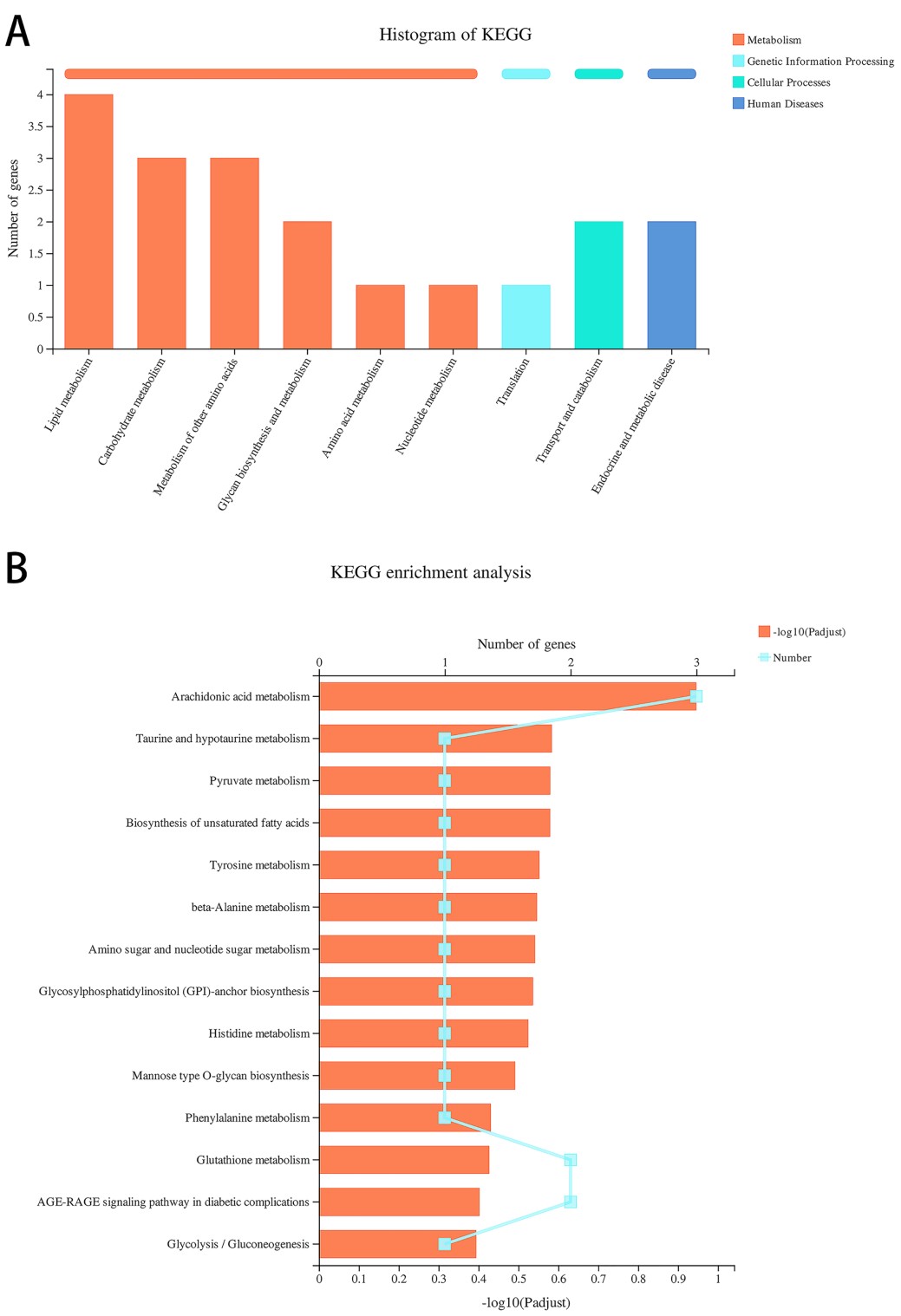

**Figure 2 Alcohol exposure regulates metabolism-related gene expression in astrocytes.** (A) KEGG annotation analysis of DEGs between the control and alcohol groups. (B) KEGG pathway analysis of DEGs between the control and alcohol groups.

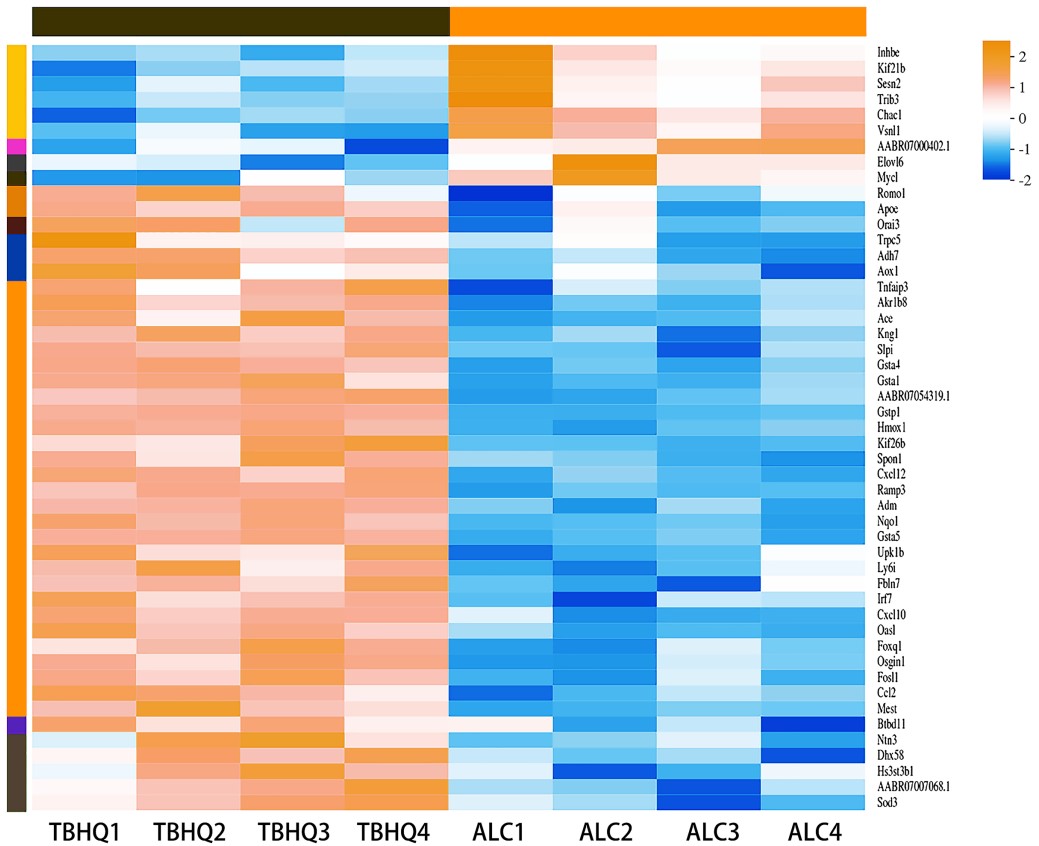

**Figure 3 Heat map of the 49 DEGs between the alcohol and TBHQ groups (*n* = 4).**

### Effects of TBHQ on alcohol-induced changes in astrocyte metabolomics—statistical analysis of differential metabolites

In total, 38 different metabolites were compared with the HMDB database, and differential metabolites were classified into six groups (Fig. 8A). Among them, lipids and lipid-like molecules, including methyl-delta-ionone, 5-isopropyl-2-(2-methyl-propyl)-2-cyclohexen-1-one, (R)-carvotanacetone, antibiotic CP 412065, humulol, melleolide L, colic acid glucuronide, lucidenolactone, PS(18:1(9Z)/0:0), LysoPC(15:0), and LysoPC(18:0), accounted for the largest proportion of metabolites. The most enriched KEGG terms related to these different metabolites was "Ether lipid metabolism" (Pathway_ID: map00565, *P* = 0.0010, Fig. 8B), including LysoPC(18:0) and 2-acetyl-1-alkyl-sn-glycero-3-phosphocholine. These results suggest that TBHQ can regulate lipid metabolic processes in alcohol-exposed astrocytes.

## DISCUSSION

To further analyze the effects of continuous alcohol intake on astrocyte function, the present study analyzed the effects of alcohol and an TBHQ on astrocyte oxidation, and the results illustrated that alcohol exposure resulted in increased MDA and decreased SOD levels in astrocytes. Meanwhile, TBHQ treatment reversed the changes in MDA and SOD

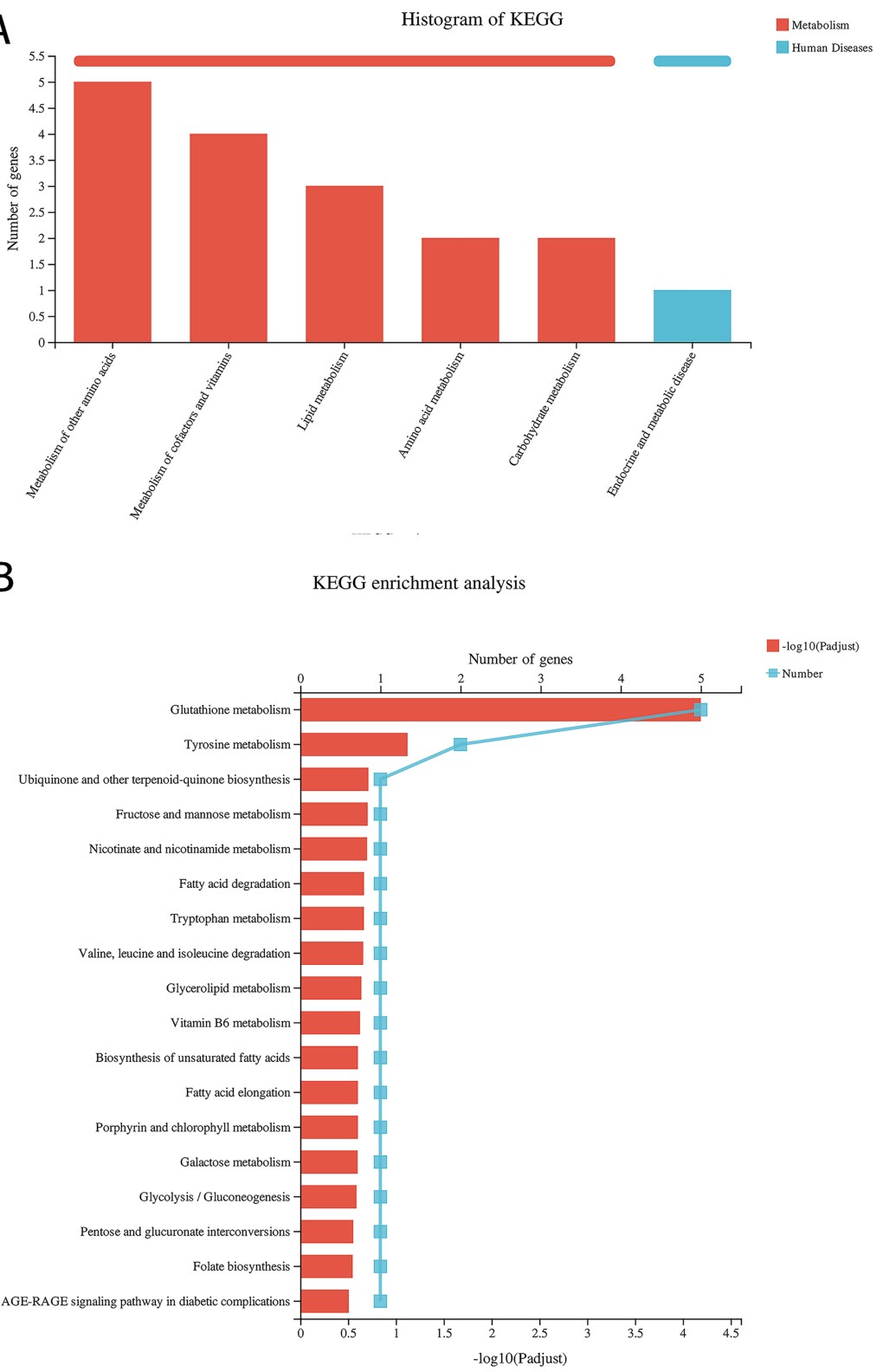

**Figure 4  TBHQ modulates the expression of genes involved in the metabolism of glutathione in astrocytes exposed to alcohol.** (A) KEGG annotation analysis of DEGs between the alcohol and TBHQ groups. (B) KEGG pathway analysis of DEGs between the alcohol and TBHQ groups.

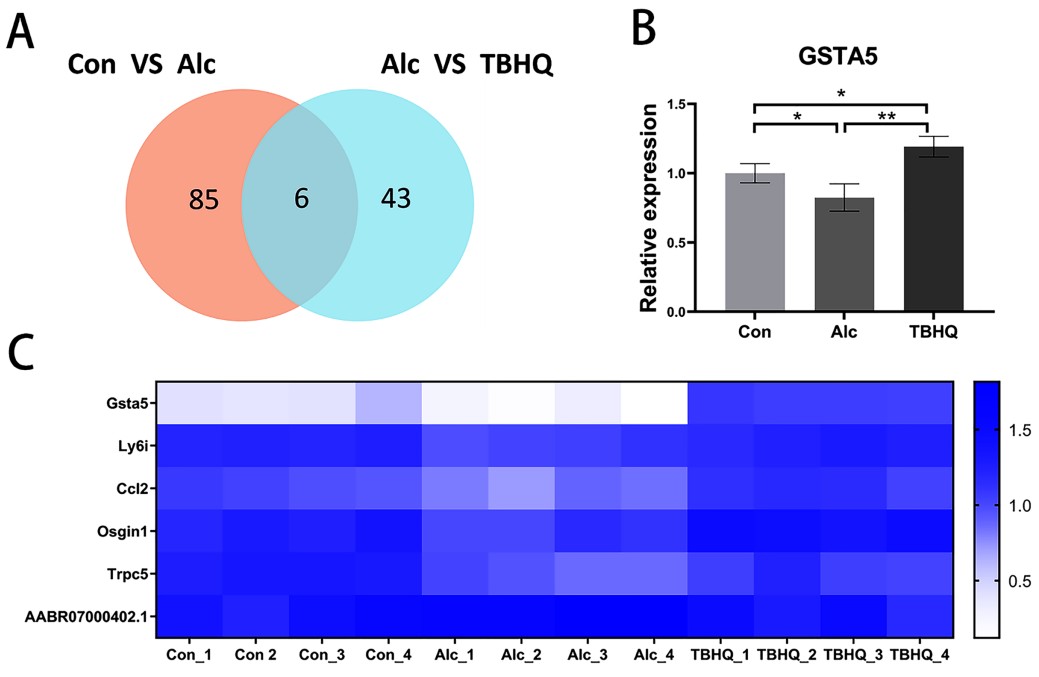

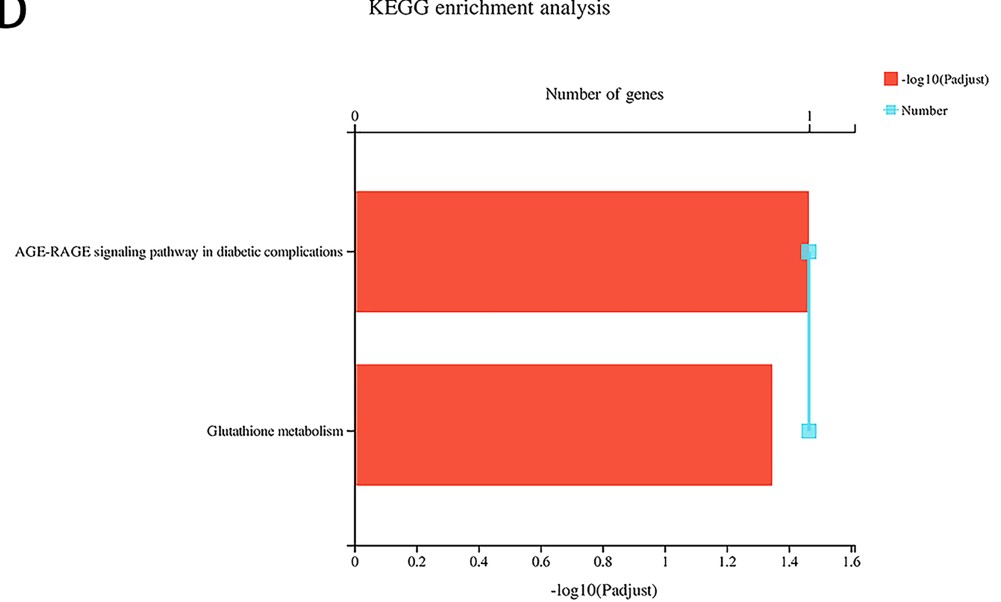

**Figure 5 Alcohol exposure decreases GSTA5 expression in astrocytes, which was reversed by TBHQ.**
(A) Venn diagram of six overlapping DEGs from the control and alcohol groups and the alcohol and
TBHQ groups. (B) Effects of alcohol and TBHQ on GSTA5 transcript levels in astrocytes ($n = 3$). (C)
Heat map of the six overlapping DEGs ($n = 4$). (D) KEGG enrichment analysis of the six overlapping
DEGs. $^*P < 0.05$, $^{**}P < 0.01$. 

levels induced by alcohol exposure in astrocytes. These results suggest that alcohol
promotes astrocyte oxidation, and these effects can be reversed by TBHQ. Alcohol is
metabolized into peroxides, and continuous alcohol intake lead to enhanced oxidation and
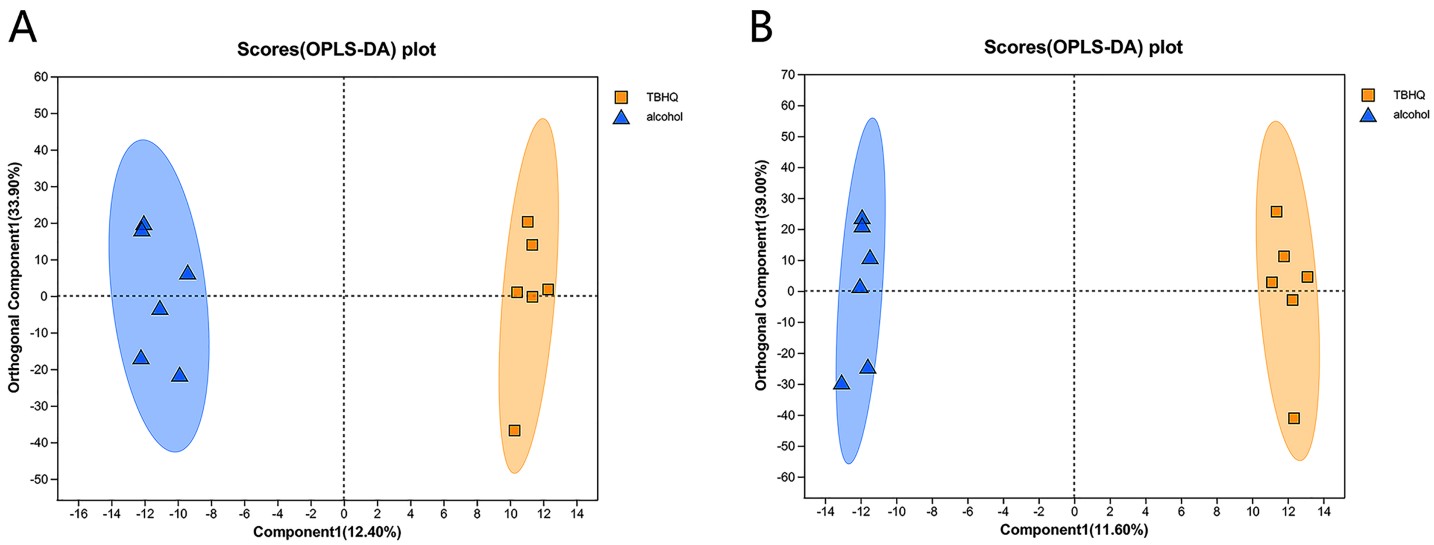

**Figure 6 OPLS-DA score plots for all detected metabolites in astrocytes treated with alcohol and TBHQ.** (A) OPLS-DA model in the cationic mode. (B) OPLS-DA model in the anionic mode.

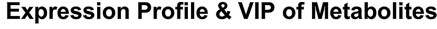

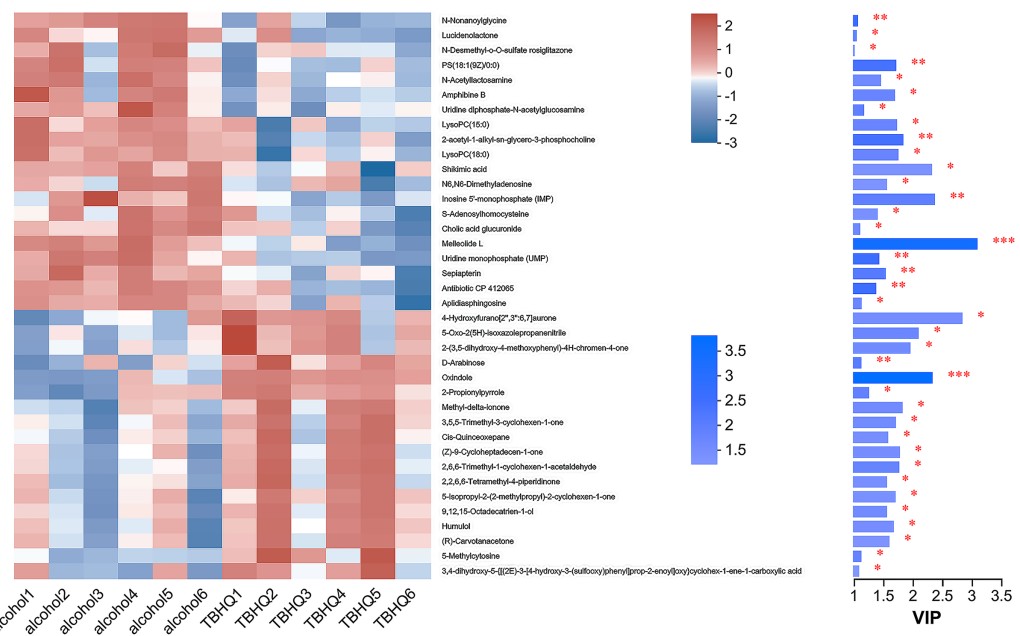

**Figure 7 Heatmap of differential metabolites between the TBHQ and alcohol groups ($n = 6$).**
$*P < 0.05$, $**P < 0.01$, $***P < 0.001$.

decreased antioxidant capacity, resulting in nerve cell damage (*Barcia et al., 2015*; *Scolaro et al., 2012*). Oxidative and antioxidant pathways play essential roles in the process of alcohol-induced brain injury (*Qin & Crews, 2012*). The results of this study illustrated that astrocytes exhibited increased oxidative levels after alcohol exposure. At the same time, treatment with TBHQ reduced oxidative levels in alcohol-exposed astrocytes.

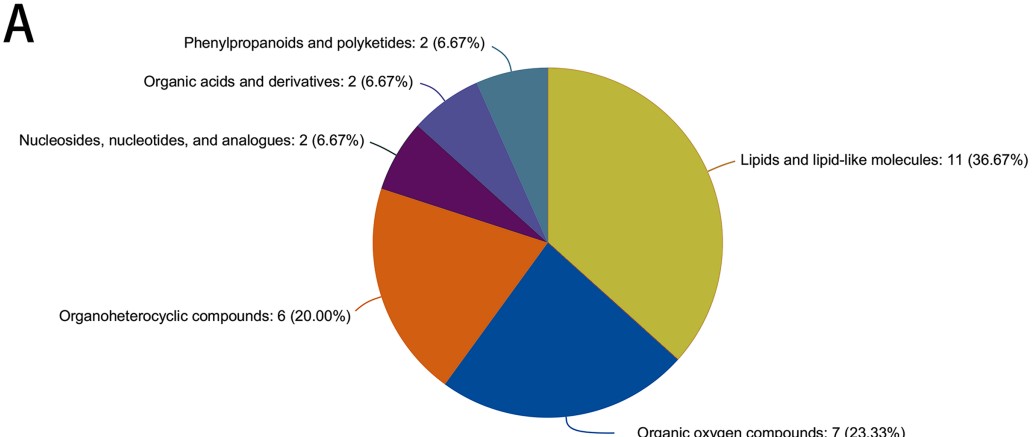

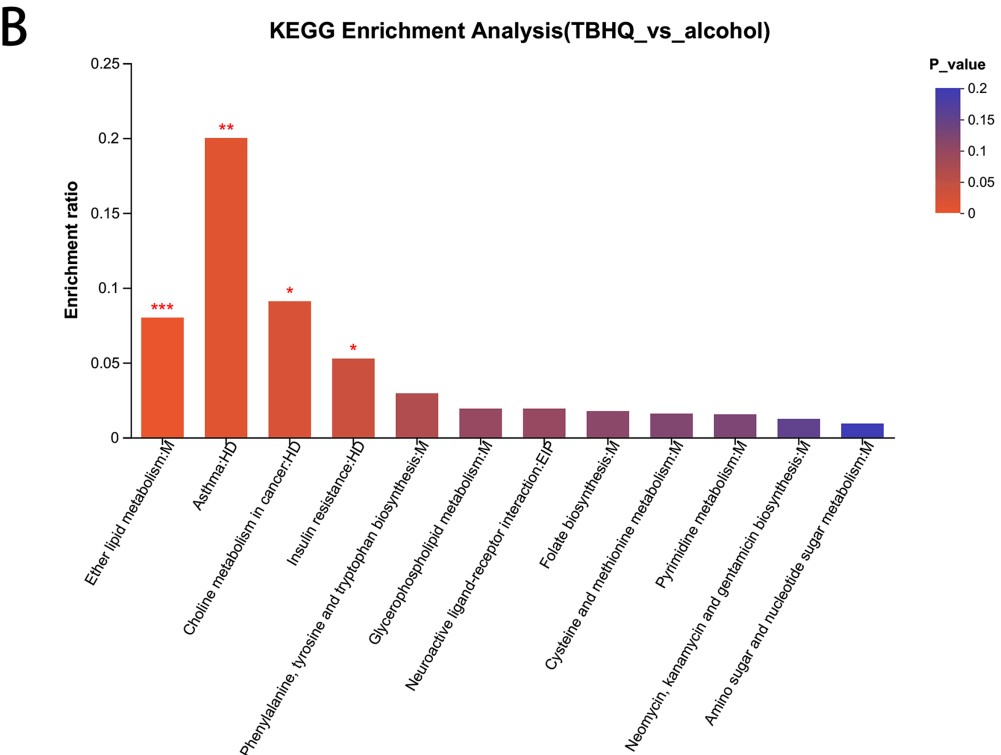

**Figure 8** **Statistical analysis of differential metabolites in astrocytes treated with alcohol and TBHQ.**
(A) Classification of differential metabolites and their proportion between the TBHQ and alcohol groups.
(B) KEGG enrichment analysis of 38 different metabolites between the TBHQ and alcohol groups.
$*P < 0.05$, $**P < 0.01$, $***P < 0.001$.

The transcription factor Nrf2 is a member of the human cap'n'collar basic region leucine zipper transcription factor family. It contains an enhancer sequence in its promoter regulatory region termed ARE, which can coordinate a multifaceted response to diverse forms of stress, enabling the maintenance of a stable internal environment (*Cuadrado et al., 2019*). Studies have revealed that Nrf2 can participate in antioxidant processes under stress conditions by promoting the expression of its downstream antioxidant genes. $LaCl_3$-

induced oxidative stress in astrocytes leads to the suppression of Nrf2 and its downstream antioxidant genes, including dehydrogenase quinone 1 (NQO1), heme oxygenase-1 (HO-1), superoxide dismutase 2 (SOD2), glutathione peroxidase 1 (GSH-Px1), and γ-glutamylcysteine synthetase (γ-GCS). However, this suppression can be counteracted by the intervention with the Nrf2 pathway agonist TBHQ, which effectively reverses these effects (*Zhang et al., 2017*). As a major metabolite of butylated hydroxyanisole, TBHQ has important antioxidant functions. It functions as an antioxidant by activating the transcription factor Nrf2 and is widely acknowledged and used as an Nrf2 pathway agonist. Studies have demonstrated that the compartment-specific impacts of TBHQ-induced Nrf2 signaling are regulated by Trx2 (*Imhoff & Hansen, 2010*). The findings from the above research collectively indicate that Nrf2 and its agonist TBHQ can effectively reduce cellular oxidation levels. The results of this study confirm that TBHQ reduce oxidative levels after alcohol exposure, suggesting that activation of the Nrf2 pathway protects against alcohol-induced damage in astrocytes through antioxidant effects. However, this result was confirmed *in vitro* and requires further validation *in vivo*.

To further analyze the mechanisms of the effects of alcohol and TBHQ on astrocyte function, transcriptome sequencing analysis was performed, which revealed that alcohol and TBHQ most strongly changed the expression of genes associated with metabolic pathways in astrocytes. Among these, TBHQ treatment and common gene analysis revealed that the most pronounced changes occurred in glutathione metabolism. Glutathione is abundant in all mammalian cells, representing the only thiol-endowed peptide with a significant role in cellular redox balance (*Bjorklund et al., 2021*). Additional studies demonstrated that activating Nrf2 signaling facilitates the transcription of downstream antioxidants, including heme oxygenase-1 and glutathione (*Xiang et al., 2022*). These results suggest that the regulation of glutathione metabolism is a potential mechanism of alcohol-induced astrocyte injury and that TBHQ protect against this damage.

In this study, both alcohol and TBHQ modulated glutathione metabolism in astrocytes. Additional studies indicated that alcohol inhibits GSTA5 expression and TBHQ reverse this process, in agreement with the transcriptome sequencing results. GSTA5 is a member of the glutathione transferase enzyme family (*Singh, Zimniak & Zimniak, 2010*). It belongs to the phase II detoxification enzyme family, and it is also involved in the regulation of the redox state of cells through different antioxidant catalytic and non-catalytic effects (*Russell & Richardson, 2023*). The active site of GSTA5 is composed of amino acid residues that can bind to substrates (such as poisons or metabolites) and transfer chemical functional groups, commonly through covalent bonds. This activity allows the enzyme to bind the substrate to glutathione, forming a glutathione covalent conjugate. GSTA5 is involved in maintaining cell integrity, resisting oxidative stress, and preventing DNA damage in the nucleus by catalyzing the binding of glutathione to various electrophilic substrates. Studies on the relationship between GSTA5 and alcohol dependence are lacking. The results of this study confirmed that alcohol exposure suppresses GSTA5 expression in astrocytes, and TBHQ promote GSTA5 expression in alcohol-exposed astrocytes. These results suggest

that the modulation of GSTA5 expression is a potential mechanism by which alcohol and TBHQ regulate glutathione metabolism and astrocyte function.

Transcriptome sequencing analysis revealed that alcohol and TBHQ produced the most significant changes in genes associated with metabolic pathways in astrocytes. To further analyze the mechanisms by which TBHQ regulate astrocyte function after alcohol exposure, this study examined untargeted metabolomics in astrocytes after TBHQ treatment and found that TBHQ could regulate the abnormal metabolism of lipids and lipid-related molecules in astrocytes after alcohol exposure, and the results revealed abnormal metabolism of ether lipid metabolism-related metabolites in astrocytes.

Ether lipids are major structural components of cell membranes that are involved in a variety of biological functions, including regulating cell differentiation, affecting cellular signaling, and reducing oxidative stress through their ability to function as potential endogenous antioxidants (*Dean & Lodhi, 2018*). It has been established that cells and model animals with abnormal ether lipid metabolism, such as plasmalogens, are more sensitive to oxidative damage. In particular, there is evidence that plasminogen can protect unsaturated membrane lipids from oxidation by singlet oxygen (*Broniec et al., 2011*). These results suggest that ether lipids are involved in cellular oxidative and anti-oxidative processes. In this experiment, two metabolites were enriched in the ether lipid metabolic pathway, namely including LysoPC(18:0) and 2-acetyl-1-alkyl-sn-glycero-3-phosphocholine. These metabolites of lipid oxidation play important roles in inflammatory response. The major lysophospholipid constituent in oxidized LDL, lysophosphatidylcholine (lysoPC), has the ability to activate p38 MAP kinase (*Chaudhuri et al., 2023*). Studies have confirmed that the amount of LysoPC(18:0) is closely related to the risk of cancer (*Kuhn et al., 2016*). 2-acetyl-1-alkyl-sn-glycero-3-phosphocholine is also known as platelet-activating factor (PAF). PAF is degraded by PAF acetylhydrolase (PAF-AH), a circulating enzyme with pro-inflammatory and anti-inflammatory activities. PAF-AH activity is considered a risk factor for coronary artery disease (*Ninio et al., 2004*). In the early stages of atherosclerosis, LDL oxidation results in the production of proinflammatory phospholipids, including platelet-activating factor (PAF) and its analogs. These findings indicate that PHF plays a role in lipid metabolism and oxidation processes. In this study, TBHQ reduced the content of alcohol-induced LysoPC(18:0) and PAF in astrocytes, indicating that TBHQs can inhibit the process of lipid oxidation, thereby reducing the production of metabolites, which could be a potential mechanism by which Nrf2 agonists regulate the function of astrocytes after alcohol exposure.

## CONCLUSION

The results of this study confirmed that alcohol exposure increased the oxidation level of astrocytes, regulated the expression of glutathione metabolism pathway genes, and reduced the expression of the GSTA5 gene, and all of these effects were reversed by the Nrf2 agonist TBHQ. These results suggest that the regulation of GSTA5 gene expression is a potential mechanism by which alcohol and Nrf2 pathway agonists regulate the oxidation level of astrocytes. Further experiments demonstrated that TBHQ could regulate ether lipid metabolism and reduce the contents of LysoPC(18:0) and PAF in astrocytes after alcohol

exposure, suggesting that the inhibition of lipid oxidation is a potential mechanism by which Nrf2 agonists protect astrocytes against alcohol-induced injury.

### Funding

This work was supported by Talent training project for basic scientific research of Heilongjiang Province Educational Commission of China (Grant numbers: 2019-KYYWF-1357) and the Jiamusi University National Fund Cultivation Program (JMSUGPZR 2022-023). The funders had no role in study design, data collection and analysis, decision to publish, or preparation of the manuscript.

### Grant Disclosures

The following grant information was disclosed by the authors:
Heilongjiang Province Educational Commission of China: 2019-KYYWF-1357.
Jiamusi University National Fund Cultivation Program: JMSUGPZR 2022-023.

### Competing Interests

The authors declare that they have no competing interests.

### Author Contributions

- Congyan Li performed the experiments, prepared figures and/or tables, authored or reviewed drafts of the article, and approved the final draft.
- Jingxin Fan performed the experiments, prepared figures and/or tables, authored or reviewed drafts of the article, and approved the final draft.
- Guangtao Sun performed the experiments, prepared figures and/or tables, authored or reviewed drafts of the article, and approved the final draft.
- Huiying Zhao performed the experiments, prepared figures and/or tables, authored or reviewed drafts of the article, and approved the final draft.
- Xiaogang Zhong analyzed the data, authored or reviewed drafts of the article, and approved the final draft.
- Xinyan Huang analyzed the data, authored or reviewed drafts of the article, and approved the final draft.
- Xiaofeng Zhu conceived and designed the experiments, authored or reviewed drafts of the article, and approved the final draft.
- Xunzhong Qi conceived and designed the experiments, authored or reviewed drafts of the article, and approved the final draft.

### Data Availability

The raw measurements are available in the Supplemental File.

## Supplemental Information

Supplemental information for this article can be found online at http://dx.doi.org/10.7717/peerj.17541#supplemental-information.

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
