# Peer review of "Nrf2 pathway activation promotes the expression of genes related to glutathione metabolism in alcohol-exposed astrocytes"

_PeerJ, doi:10.7717/peerj.17541_

## Round 0.1 · original submission · Major Revisions

While reviewers appreciated the manuscript, they also highlighted several major issues. Please address them fully to plan to submit a revised manuscript.

Reviewer 1 ·

Basic reporting

The authors provide an original study related to the role of Nrf2 in alcohol-exposed astrocytes, finding that the antioxidant response could be mediated by the glutathione’s metabolism. The manuscript need improvements, particularly in the area of discussion. The authors must improve at least the following points:


The correct form for NF-E2–related factor 2 is Nrf2 not “NRF2”.
In abstract:
Explain briefly why LysoPC(18:0) and PAF are important (also, take care of the acronyms and explain correctly).
Introduction:
Line 90: what do the authors want to say with … “the number of astrocytes is greatly sensitive to alcohol”?
Line 102: define TBHQ. The authors mention previous studies but they do not explain them and do not reference them. Also, the explanation of TBHQ is very loose and needs to be reinforced.
Methodologies
In qRT-PCR the authors should mention the equipment used, also the software that they utilized.
In statistical analysis, the authors should inform the statistic software (version and year), also they must inform the number of replicates and if they used media or average.
Results
The authors should inform “the percentage or times related” to increased or reduced activities shown.
Legends of figures must be better developed. Figures 2B and 3A results impossible to read the names.
Discussion
Line 251: Previous studies ??? the authors did not mention them.
Line 252: “agonist” which??? The authors should improve the discussion to better understand what they want to explain.
Line 256 to 259: they must work and deepen and improve the ideas (subtracted proteins?? Antioxidants? Enzymes?)
Lien 259 to 261: inform that the assay was in vitro and the limitations of this related to the results.
The authors should discuss the activity of TBHQ, its limitations, and its relation with antioxidant pathways (mechanism of action).
Line 263 to 268: There is no type of discussion in the paragraph.
The discussion in general is quite weak, it must be improved it.

Experimental design

No comment

Validity of the findings

The authors should better explain how they performed and validated the statistical data. They must also report the number of replicates, samples and others tested.

Additional comments

The discussion in general is quite weak, it must be improved it.

Reviewer 2 ·

Basic reporting

In this manuscript, Li et al. explored the effects of alcohol and the NRF2 agonist TBHQ on astrocyte function, gene expression, and metabolite content. The results of this study indicate that TBHQ reverses alcohol-induced oxidative stress by modulating glutathione and ether lipid metabolism, suggesting the potential of NRF2 agonists in protecting astrocytes from alcohol-related damage. The topic of this work aligns well with the interests of PeeJ readers. I recommend the acceptance of this work, subject to minor revisions. Please find my comments below:
Major Points:
1. The manuscript's discussion of the NRF2 pathway is hard to follow. I suggest adding a schematic diagram of the metabolic pathways to aid readers' comprehension.
2. In line 116, the "Alc" group's alcohol concentration is stated as "75 mM". Using mM is uncommon for describing alcohol concentration; I advise converting this to percent concentration (% w/v).
3. Following the previous point regarding the concentration, why was 75 mM selected? Please provide a reference. Adding data for different concentrations would robustly support the study's conclusions.
4. The abbreviation TBHQ is not defined; I presume it refers to tert-Butylhydroquinone. I would recommend to discuss the mechanisms of this inhibitor somewhere in the paper.
5. Adding another schematic to summarize the paper's findings on metabolic pathways would be recommended.
Minor Points:
1. Overall, the figures are of poor quality, making it difficult to discern the details. Enhancing the quality of all figures and enlarging the font is necessary.
2. The error bars in Figure 1 are incomplete at the bottom.
3. In line 179, the value "RNO00590, P=0.000" appears questionable. Please confirm the number.

Experimental design

Discussed in basic reporting

Validity of the findings

Discussed in basic reporting

Additional comments

Discussed in basic reporting

·

Basic reporting

This manuscript presents an investigation into the role of NRF2 pathway activation in astrocytes exposed to alcohol, focusing on its effects on glutathione metabolism. The research is well-structured, adhering to scientific standards with a clear division into introduction, methods, results, and discussion sections. The study is based on the hypothesis that NRF2 pathway activation through an agonist (TBHQ) could mitigate the oxidative stress induced by alcohol in astrocytes, specifically by modulating the expression of genes related to glutathione metabolism.

Experimental design

The methods section is comprehensive, detailing cell culture and treatment processes, transcriptome sequencing, qRT-PCR analysis, LC-MS/MS, and metabolomics data analysis. However, please define the concentration of alcohol used in % of EtOH. Please also include the sample size for figure 1A &B so that we could interpret the p-value associated with it.

Validity of the findings

Key findings include:
Alcohol exposure increased MDA levels and decreased SOD levels in astrocytes, indicating oxidative stress.
TBHQ treatment reversed these effects, suggesting a protective role against oxidative damage.
Transcriptome and metabolomics analyses showed that TBHQ modulated the expression of genes involved in glutathione metabolism and ether lipid metabolism, highlighting potential protective mechanisms.
The study concludes that NRF2 pathway activation via TBHQ can mitigate alcohol-induced oxidative stress in astrocytes by modulating glutathione metabolism, suggesting a potential therapeutic avenue for alcohol-related brain injury.

It is nice to see the qRT-PCR result confirms the RNA seq result for one example gene, but it would be better to include another two examples.

Additional comments

Please make the figure axis labelling bigger for figure 2 for better visualization
Please make the figure axis labelling bigger for figure 3 for better visualization
Please make the figure axis labelling bigger for figure 5 for better visualization
Please make the figure axis labelling bigger for figure 6 for better visualization

---

## Round 0.2 · Minor Revisions

Regrettably, I must inform you that your manuscript has not been accepted for publication yet. Several critical issues have been identified that require substantial revision before the manuscript can be considered further.

Firstly, numerous errors in spacing and language were noted throughout the manuscript, which significantly detracts from its readability and professionalism. Additionally, the Methods and Materials section lacks specificity and completeness. For instance, despite the authors' response to a reviewer regarding qPCR analysis details, this crucial information has not been incorporated into the manuscript. It's imperative to provide all necessary details, including technical equipment, kits' catalog numbers, and antibodies, to ensure reproducibility by other researchers.
Moreover, the manuscript contains redundant statements without proper referencing and background setting, which renders certain statements nonsensical. The repetition of a sentence without citation raises concerns about the manuscript's coherence and academic integrity. Therefore, thorough revision and proofreading are warranted to address these issues.
Additionally, there are shortcomings in the presentation of results. The abstract should highlight significant values and results rather than providing a broad description. Furthermore, figure legends lack clarity and fail to adequately explain the content and significance of the images. Ambiguities in units and labeling also need to be rectified to ensure clarity and precision.
Furthermore, the manuscript lacks a comprehensive discussion of the relevant literature and clinical outcomes related to the topic. While the authors mention the role of the Nrf2 pathway in alcohol-induced brain injury, they fail to provide sufficient context or discuss previous research findings. It's essential to critically evaluate the existing literature and clearly articulate the novelty and significance of the current study in this context.
Regarding the provided research on Nrf2 pathway activation in alcohol-exposed astrocytes, while the results demonstrate promising findings, there are concerns about the adequacy of in vitro work performed and the quality of figures presented. It's crucial to ensure that all figures meet the minimum quality standard and resolution requirements.

In summary, while the manuscript addresses an important topic, significant revisions are necessary to address the identified issues adequately before acceptance.

Reviewer 2 ·

Basic reporting

The revised manuscript addressed all my concerns and its current form is acceptable.

Experimental design

no comments

Validity of the findings

no comments

·

Basic reporting

I have no further comments.

Experimental design

I have no further comments.

Validity of the findings

I have no further comments.

Additional comments

I have no further comments.

---

## Round 0.3 · accepted · Accept

Dear authors, many thanks for your submission and work throughout this process. I believe that your work can now be accepted for publication and presents very interesting data. The best of luck for your future endeavors.